# Divergent landscapes of A-to-I editing in postmortem and living human brain

Miguel Rodriguez de los Santos[1,2], Brian H. Kopell[1,2], Ariela Buxbaum Grice[1], Gauri Ganesh[1], Andy Yang[1], Pardis Amini[1], Lora E. Liharska[1], Eric Vornholt[1], John F. Fullard [1], Pengfei Dong [1], Eric Park[1], Sarah Zipkowitz[1], Deepak A. Kaji[1], Ryan C. Thompson [1], Donjing Liu[1], You Jeong Park[1], Esther Cheng [1], Kimia Ziafat[1], Emily Moya [1], Brian Fennessy [1], Lillian Wilkins[1], Hannah Silk[1], Lisa M. Linares[1], Brendan Sullivan[1], Vanessa Cohen[1], Prashant Kota[1], Claudia Feng [1], Jessica S. Johnson[1], Marysia-Kolbe Rieder[1], Joseph Scarpa[1], Girish N. Nadkarni [1], Minghui Wang [1], Bin Zhang[1], Pamela Sklar[1], Noam D. Beckmann [1], Eric E. Schadt[1], Panos Roussos [1,2], Alexander W. Charney [1,2] & Michael S. Breen [1,2] ✉

Adenosine-to-inosine (A-to-I) editing is a prevalent post-transcriptional RNA modification within the brain. Yet, most research has relied on postmortem samples, assuming it is an accurate representation of RNA biology in the living brain. We challenge this assumption by comparing A-to-I editing between postmortem and living prefrontal cortical tissues. Major differences were found, with over 70,000 A-to-I sites showing higher editing levels in postmortem tissues. Increased A-to-I editing in postmortem tissues is linked to higher *ADAR* and *ADARB1* expression, is more pronounced in non-neuronal cells, and indicative of postmortem activation of inflammation and hypoxia. Higher A-to-I editing in living tissues marks sites that are evolutionarily preserved, synaptic, developmentally timed, and disrupted in neurological conditions. Common genetic variants were also found to differentially affect A-to-I editing levels in living versus postmortem tissues. Collectively, these discoveries offer more nuanced and accurate insights into the regulatory mechanisms of RNA editing in the human brain.

Post-transcriptional RNA modifications play important roles in complex functions of the central nervous system (CNS)[1,2]. The conversion of adenosine nucleosides to inosine (A-to-I) represents one of the most abundant RNA modifications cataloged in the human brain[2–5]. Inosine is recognized as guanosine (G) upon translation or sequencing, thus, the net effect of A-to-I editing is a post-transcriptional A-to-G transition. A family of three adenosine deaminases acting on RNA (ADAR) enzymes drives these conversions on RNA transcripts, which underlie diverse molecular functions. ADAR1 (*ADAR*) exists in two isoforms, P110 and P150, both of which are responsible for A-to-I editing along endogenous long double-stranded RNA. Notably, only the P150 isoform is inducible by interferon, which leads to an upregulation of A-to-I editing activity and plays a crucial role in modulating innate immune responses[6–8]. ADAR2 (*ADARB1*) is expressed in the CNS and is recognized for its role in editing protein-coding sequences, which results in the production of functionally diverse protein isoforms crucial for typical neurodevelopment[9–11]. ADAR2 is also capable of editing sites within SINE elements[11]. ADAR3 (*ADARB2*) is expressed exclusively in the brain but cannot catalyze A-to-I and is proposed to be a negative regulator of A-to-I editing[12,13]. In the mammalian brain, thousands of

[1]Icahn School of Medicine at Mount Sinai, New York, NY 10029, USA. [2]These authors contributed equally: Miguel Rodriguez de los Santos, Brian H. Kopell, Panos Roussos, Alexander W. Charney, Michael S. Breen. ✉e-mail: michael.breen@mssm.edu

highly regulated A-to-I editing sites have been discovered across anatomical regions and cell types[13–15], as well as neuronal maturation and brain development[16–18]. Aberrant regulation of A-to-I editing in the brain has also been linked to the etiology of several neurological disorders[19–23], further underscoring the physiological significance of A-to-I editing in the CNS. Yet, as an understanding of A-to-I editing in the CNS burgeons, it is important to acknowledge that these advancements are exclusively driven by studies in postmortem tissues.

The brain is highly vulnerable to changes in blood flow and oxygen levels, and mammalian cells require oxygen to maintain cellular and tissue viability[24]. Shortly after death, intracellular acidosis and oedema elicit secondary injury to membranes and organelles, causing an irreversible cascade of apoptosis, necrosis, and axonal damage. This is followed by activation of innate immune responses, leading to end-organ injury and widespread metabolic acidosis[24–27], which could alter ADAR and A-to-I editing. Moreover, DNA is relatively stable over extended postmortem periods, RNA is much more chemically labile and sensitive[28]. To this end, we postulate that molecular responses to ischemic exposures, and the contribution of postmortem-induced innate immune responses, likely alter the landscape of A-to-I editing in postmortem brain tissues, skewing a comprehensive biological understanding of RNA editing in the CNS. Indeed, a supportive, initial report has suggested increased *Alu* editing in non-CNS postmortem tissues compared to tissues obtained from ventilator-dependent donors[29]. Given these considerations, it is imperative to underscore the significance of distinguishing between postmortem and living CNS tissues, particularly as RNA editing studies of the brain play an increasingly pivotal role in advancing our knowledge of brain aging and disease.

To address this, the current study investigates the fundamental differences of A-to-I editing between postmortem and living human dorsolateral prefrontal cortex (DLPFC) tissues (Fig. 1). We anchor our investigation around the state-of-the-art Living Brain Project (LBP)[30], whereby DLPFC tissues from living people were obtained during neurosurgical procedures for deep brain stimulation (DBS) an elective treatment for neurological illnesses. For comparison, a cohort of postmortem DLPFC tissues across three brain banks was assembled to match the living cohort to the extent possible for key demographic and clinical variables (see full cohort description in Supplemental Data 1). All samples underwent joint genomic data generation. We leveraged paired whole-genome sequencing (WGS) and bulk-tissue RNA-sequencing of the DLPFC from 164 living participants, including 78 with unilateral biopsies and 86 with bilateral biopsies, as well as 233 partially matched postmortem tissues. A non-overlapping, independent collection of 31 living and 21 postmortem DLPFC tissues also underwent single nuclei RNA-sequencing (snRNA-seq). Herein, we provide substantial evidence for significant differences in A-to-I editing profiles between postmortem and living human brain tissues, which are more evident in non-neuronal cell types (Fig. 1). Our results underscore that while the investigation of fresh brain tissues can enhance understanding of RNA editing biology, they also provide missing context and support for the utility of postmortem brain tissues in A-to-I editing studies relevant to human brain health and disease.

## Results

**Global *Alu* editing is elevated in postmortem prefrontal cortex**
Since most A-to-I editing occurs in *Alu* elements[3,4], we first computed an *Alu* editing index (AEI) for each sample. The AEI is quantified by measuring the total number of edited adenosines over all adenosines with supporting RNA-sequencing read coverage in *Alu* elements across the entire transcriptome and is a metric of global *Alu* editing activity (see Materials and Methods) (Supplementary Data 1). A significant increase in the AEI was observed in postmortem relative to living DLPFC ($p = 4.3 \times 10^{-75}$; Cohen's $d = 2.88$) (Fig. 2A). A transcriptome-wide comparative analysis compared postmortem to living DLPFC revealed

that this shift was accompanied by heightened expression of *ADAR* ($q$ value = $9.3 \times 10^{-87}$), *ADARB1* ($q$ value = $3.5 \times 10^{-32}$), and *ADARB2* ($q$ value = $2.5 \times 10^{-21}$) in postmortem DLPFC (Fig. 2B). Notably, *ADAR* was the 15th most differentially expressed gene in postmortem DLPFC and was strongly correlated with the AEI ($r = 0.65$) (Fig. S1). Next, a linear mixed model quantified the fraction of global *Alu* editing variance explained by known biological and technical factors. Differences between living and postmortem tissues explained the largest amount of *Alu* editing variability (~72%) (Fig. 2C), while minimal variance was explained by other known factors, including differences by medical diagnosis (<0.5%), brain banks (<0.5%) and estimated neuronal cell type proportions (<0.5%) (Fig. S2), extended postmortem interval (PMI; <0.5%) and RNA integrity (RIN; <0.5%).

To further explore the possible influence of differences related to PMI and RNA degradation on *Alu* editing in living and postmortem tissues, two supporting analyses were conducted. First, the minimal effect of extended PMI on the AEI was validated by studying 2841 independent transcriptome samples across four large-scale postmortem brain consortia (GTEx project [$n = 1129$]; Mount Sinai Brain Bank [$n = 876$], PsychENCODE [$n = 251$]; BrainSpan [$n = 585$]). These secondary postmortem analyses confirmed weak associations between PMI and the AEI ($r^2 = 0.006$, $r^2 = 0.03$, $r^2 = -0.007$, $r^2 = -0.019$, respectively) (Fig. S3), indicating that elevated global *Alu* editing in postmortem tissue is not likely driven by extended PMI. Second, we measured A-to-I editing dynamics using an existing molecular degradation assay of the human DLPFC[31], whereby no significant changes in *ADAR* expression nor the AEI were observed throughout the increasing degradation stages of the DLPFC (Fig. S4). These results suggest that potential differences in PMI and RNA degradation do not fully account for the observed changes between postmortem and living DLPFC.

We next replicated features of RNA editing across 206,568 single nuclei sequenced from a non-overlapping sample of 31 living and 21 postmortem DLPFC tissues (Supplemental Data 1). Pseudo-bulk snRNA-seq pools were used to confirm heightened global *Alu* editing levels ($p = 5.5 \times 10^{-7}$) as well as increased mRNA expression per cell for *ADAR* ($p = 0.01$), *ADARB1* ($p = 0.0004$), and *ADARB2* ($p = 0.02$) in postmortem relative to living DLPFC (Fig. S5A, B) (see Materials and Methods). Unsupervised dimensionality reduction applied to all data identified nine discrete cell type clusters (Fig. 2D). Fewer oligodendrocytes were detected in postmortem DLPFC (OLI, $p = 1.8 \times 10^{-7}$), while fewer excitatory neurons (EXC1, $p = 1.8 \times 10^{-7}$) and inhibitory neurons (INT1, $p = 1.5 \times 10^{-9}$) were detected in living DLPFC (Fig. 2E). *ADAR* was ubiquitously expressed across all cellular populations, *ADARB1* was expressed in a subset of inhibitory and excitatory neurons, and *ADARB2* was uniquely expressed in oligodendrocytes and a small subset of inhibitory cells (Fig. 2F). Pseudo-bulk pools were generated for each cell type per donor (Fig. S5C) and used to compare the expression of ADAR enzymes and the AEI between living and postmortem DLPFC within each cellular population. Using this approach, most cell populations in postmortem DLPFC displayed significantly increased expression of *ADAR, ADARB1,* and *ADARB2* (Fig. 2G) together with an increased AEI relative to living DLPFC (Fig. 2H). Notably, the largest cell type increases in global *Alu* editing in postmortem DLPFC occurred in microglia, endothelial cells (ENDO), and oligodendrocyte precursor cells (OPCs) (Fig. 2H). These increases were concordant with increased expression of *ADAR* ($r = 0.31$), *ADARB1* ($r = 0.57$) (Fig. 2I), and less so *ADARB2* ($r = 0.09$) (Fig. S5F).

## Accurate detection of A-to-I sites in living and postmortem cortex
To study individual sites underlying these global changes, we cataloged high-confidence RNA sites using two complementary site calling techniques followed by a series of comprehensive detection-based thresholds to safeguard against false positives (see Materials and

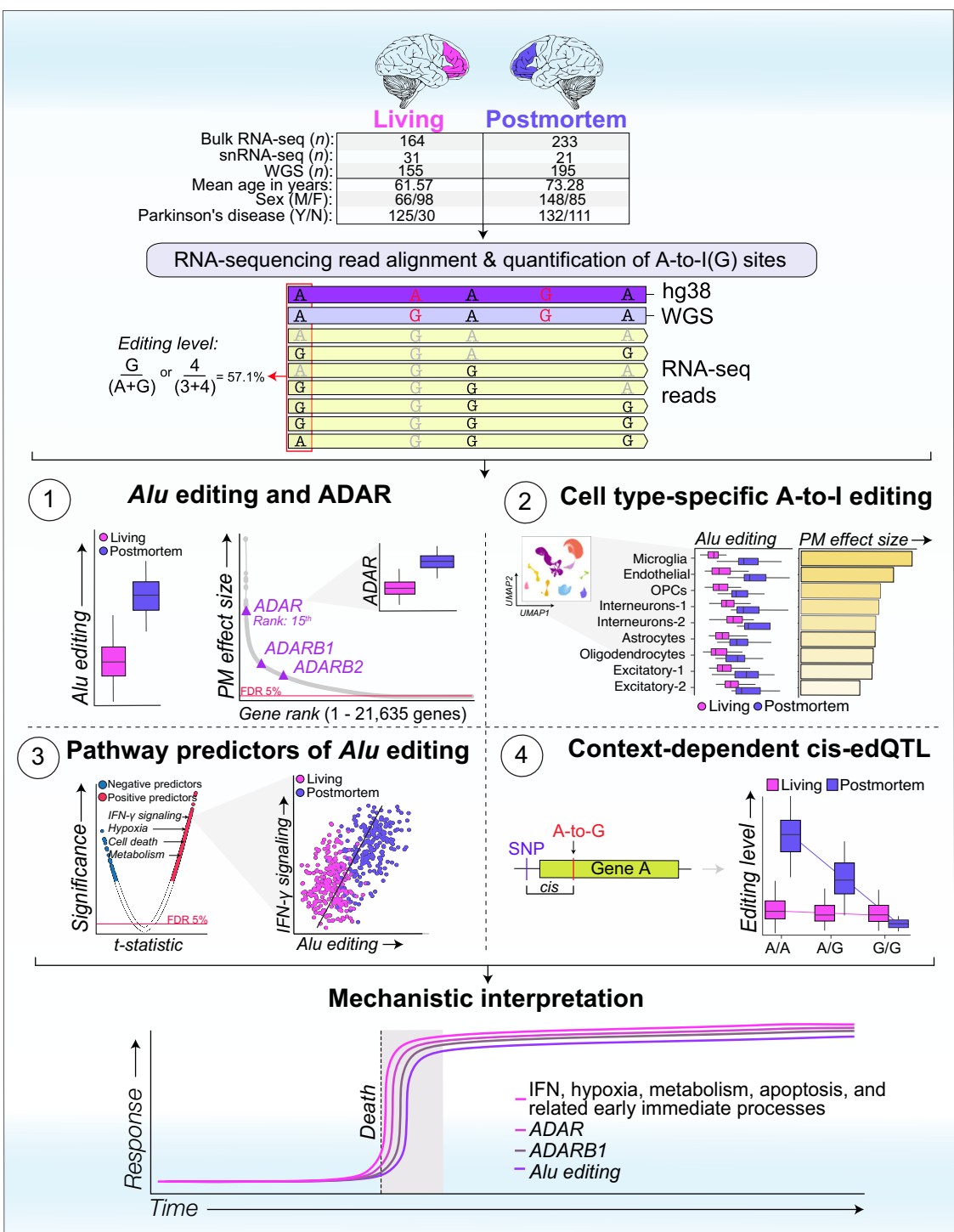

**Fig. 1 | Overview of study design and multi-omic utilization.** This study leverages a comprehensive set of multi-omic data from the Living Brain Project, including: (i) Bulk RNA-sequencing data from 164 living and 233 postmortem dorsolateral prefrontal cortex (DLPFC) samples; (ii) Single-nuclei RNA-sequencing data from an independent subset of 31 living and 21 postmortem DLPFC samples, ensuring no participant overlap with the bulk sequencing cohort; and (iii) Paired whole-genome sequencing (WGS) data from 155 living and 195 postmortem DLPFC samples. Detailed cohort demographics are detailed in Supplemental Data 1. Comprehensive analyses were conducted to quantify global *Alu* editing levels and individual A-to-I editing sites, with subsequent investigations encompassing bulk tissue comparisons, cell type-specific editing patterns, pathway-driven predictors of editing, and the genetic influences on A-to-I editing dynamics. We propose a mechanistic model to frame the interpretation of our overall findings.

Methods). Here, a de novo caller was used to uncover high-quality A-to-I sites not already cataloged in existing databases, together with a supervised approach applied to three large lists of known sites (Fig. S6A, Supplemental Data 1). A mean of 193,195 editing sites were detected per sample in living DLPFC, and 295,343 sites were detected per sample across postmortem tissues. Importantly, these sites showed hallmark characteristics of ADAR-mediated RNA editing, as the majority of sites: (1) were A-to-I sites (~93% living, ~95% postmortem); (2) mapped to *Alu* elements (~82% living, ~83% postmortem) (Fig. S6A); (3) were predominantly known sites cataloged in editing databases

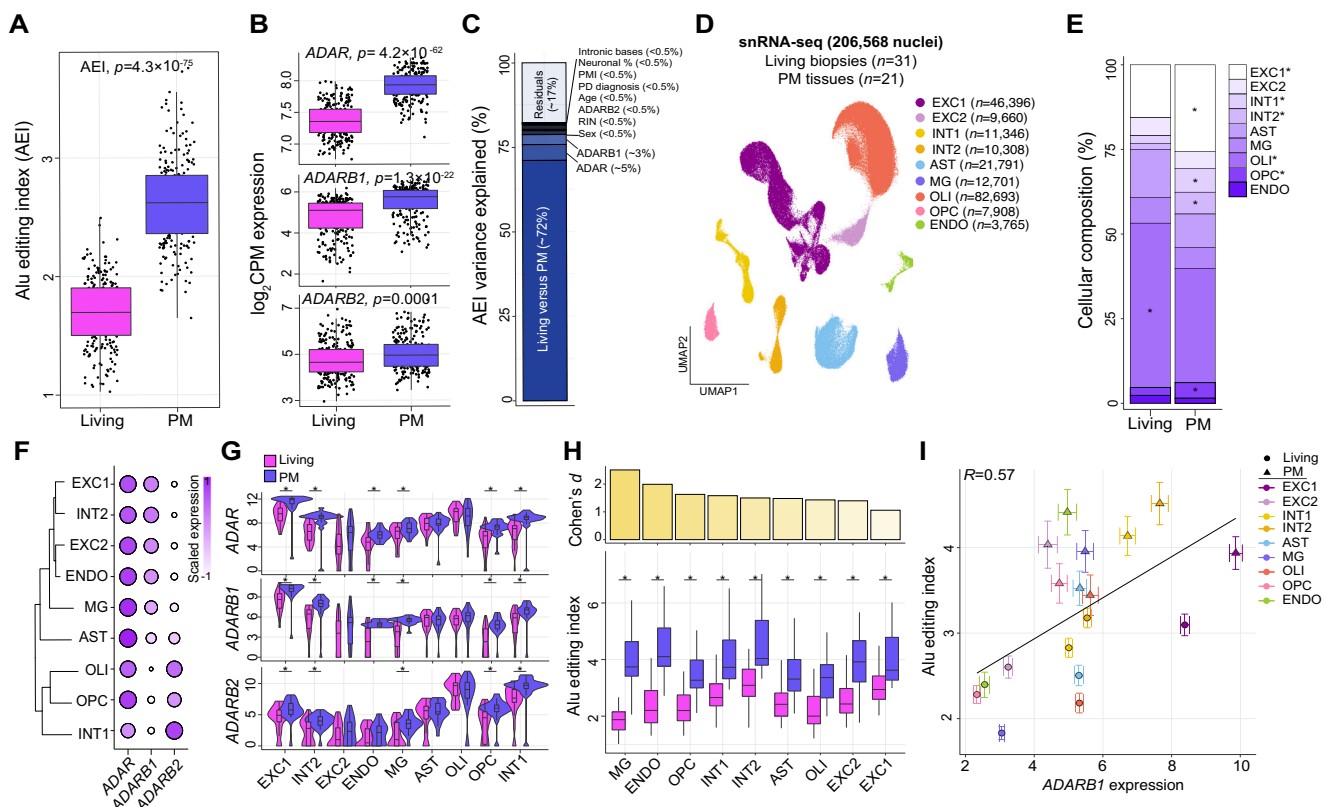

**Fig. 2 | Global *Alu* editing across living and postmortem DLPFC. A** *Alu* editing index (AEI; *y* axis) computed on bulk RNA-seq from living and postmortem (PM) DLPFC. Two-sided linear regression was used to test for significance. **B** *ADAR*, *ADARB1,* and *ADARB2* normalized expression profiles on bulk including RNA-seq between living and PM. All boxplots show the medians (horizontal lines), upper and lower quartiles (inner box edges), and 1.5× the interquartile range (whiskers). Reported BH adjusted *p* values were derived from a moderated *t* test comparing transcriptome-wide gene expression between living and postmortem tissue. **C** Linear mixed model explaining AEI variance by eleven known biological and technical factors. **D** UMAP dimension reduction analysis of snRNA-seq classified nine unique cell populations. Values in brackets indicate the number of cells per sub-population: excitatory (EXC) and inhibitory (INT) neurons, astrocytes (AST), microglia (MG), oligodendrocytes (OLI), OLI precursor cells (OPCs), endothelial cells (ENDO). **E** The mean frequencies for each cell population quantified between living and PM. Two-sided linear regression was used to test for significance. (**F**) Hierarchical clustering of scaled *ADAR*, *ADARB1,* and *ADARB2* expression across all

cell populations. **G** Cell type-specific *ADAR*, *ADARB1,* and *ADARB2* expression for living and PM. **H** *Alu* editing index computed for each cell population for each donor and compared across living and PM samples (bottom). PM-induced effect sizes calculated by Cohen's d for each cell population (top). **G, H** Two-sided linear regression was used to test for significance (˙denotes *p* < 0.05). All boxplots show the medians (horizontal lines), upper and lower quartiles (inner box edges), and 1.5× the interquartile range (whiskers). Two-sided linear regression was used to test for significance. **I** Pearson's correlation coefficient between the mean *Alu* editing index and mean normalized *ADARB1* expression for each cell population according to living and PM samples. Standard error bars capture group-wise variance within living and postmortem tissues, respectively. RNA-seq analysis encompassed 164 and 233 biologically independent samples from living and postmortem sources, respectively. Single-nucleus RNA-seq was conducted on 31 living and 21 PM biologically independent samples. All boxplots in this figure show the medians (horizontal lines), upper and lower quartiles (inner box edges), and 1.5 × the interquartile range (whiskers).

(~85% living, ~84% postmortem) (Supplemental Data 1); (4) were sites with low editing levels (20–40%; Fig. S6B); and (5) commonly mapped to introns and 3′ UTRs (Fig. S6C) (Supplemental Data 1). Further, while site discovery was largely correlative with sequencing depth, no major differences in library depth were observed between living and postmortem DLPFC (Fig. S6D–F), suggesting it is not a driver for the observed differences. Subsequent analyses examined exclusively A-to-I sites.

### Highly dynamic A-to-I editing and RNA recoding between living and postmortem DLPFC

Two approaches probed RNA editing differences between living and postmortem DLPFC. First, analysis of A-to-I sites detected at significantly different frequencies between postmortem and living DLPFC identified 28,417 sites preferentially enriched in postmortem tissues and 1436 sites enriched in living tissues, the majority in intronic regions (Fig. S7A, Supplemental Data 2). Enrichment patterns were not explained by changes in gene expression levels (Fig. S7B). Second, A-to-I sites with significantly different mean editing levels were queried

between living and postmortem DLPFC, focusing analyses on 54,825 A-to-I sites found across all samples in this study. Principal component analysis (PCA) distinguished living from postmortem tissues based on editing levels for these sites and PC1 was strongly correlated with differences between these two groups ($r = 0.92$, $p = 9.3 \times 10^{-163}$) (Fig. 3A). Statistical analysis identified differentially edited sites: 41,044 showed higher editing in postmortem tissue ("postmortem-biased") and 1449 showed higher editing in living tissue ("living-biased") (Fig. 3B, Supplemental Data 2). Sites annotated as postmortem-biased were edited ~37% within the living DLPFC sample (Fig. 3C). Both postmortem- and living-biased sites predominantly mapped to non-coding regions, with a moderate enrichment of living-biased sites mapping to 3′UTRs and coding regions (Fig. 3D). Overall, editing levels for these sites were most strongly correlated with increased expression of *ADAR* (mean $r = 0.29$) in postmortem DLPFC followed by changes *ADARB1* (mean $r = 0.18$), with little effect explained by *ADARB2* (mean $r = 0.05$) (Fig. 3E).

A small fraction of differentially edited sites was cataloged as RNA recoding sites (~0.13%, $n = 58$ sites), which introduce nonsynonymous

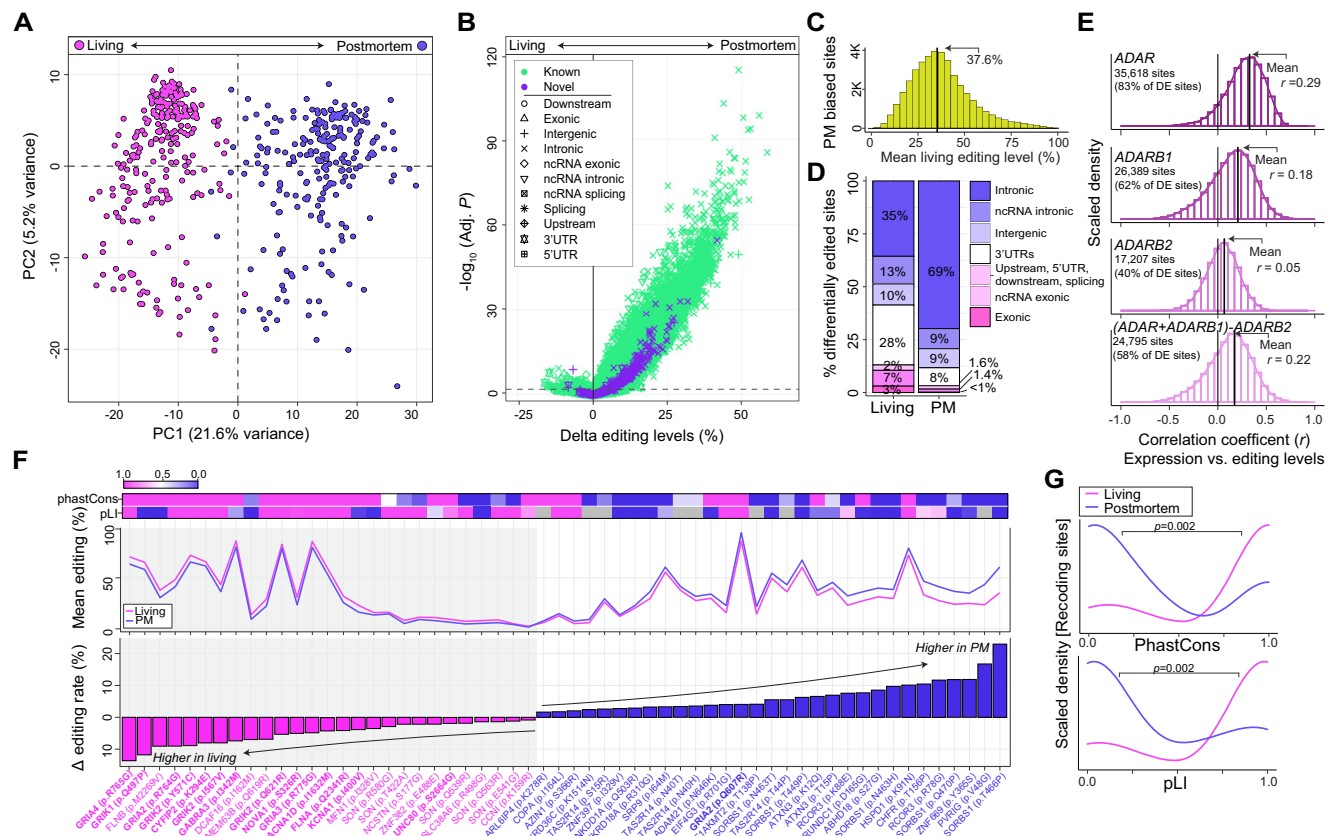

**Fig. 3 | Dynamically regulated A-to-I sites between living and postmortem DLPFC. A** Principal component analysis of editing levels for 54,825 high-confidence sites detected across all samples in the current study. **B** Differential editing analysis compare delta editing levels (%; *x* axis) and strength of significance ($-\log_{10}$ adjusted *P*, *y* axis) for each site between living and post-mortem (PM) DLPFC. Sites are colored by novelty (i.e., detection in REDIportal) and shaped uniquely by genic region. Reported BH adjusted *p* values were derived from a moderated *t* test comparing transcriptome-wide A-to-I editing levels between living and postmortem tissue. **C** Frequency distribution of mean editing levels in living DLPFC (*x* axis) based on PM biased sites (*y* axis). **D** The fraction of genic regions for all living biased and PM biased sites. **E** Frequency distributions of Pearson's correlation coefficients (*x* axis) between the expression for *ADAR*, *ADARB1*, *ADARB2* relative to editing levels for 54,825 sites. An additional analysis

modeled *ADAR* + *ADARB1-ADARB2* to capture *ADAR* and *ADARB1* effects. The total number of sites with significant correlations are listed in the top right corner of each histogram. **F** Dynamic recoding sites: 27 living-biased recoding sites and 31 postmortem-biased recoding sites ordered by their effect size differences (*y* axis, lower) and plotted alongside with the mean editing levels (*y* axis middle). The strength of evolutionary conservation (phastCons) was measured for each site and the probability of being loss of function intolerant (pLI) was measured for each gene (top). **G** Frequency distributions demonstrating that living biased recoding sites are often more strongly evolutionarily conserved and map to genes with higher pLI relative to PM biased recoding sites. Mann Whitney *U* test was used to test for significance. Living Brain Project data encompassed 164 and 233 biologically independent samples from living and postmortem sources, respectively.

substitutions in protein-coding regions (Fig. 3F). Ranking these recoding sites by their effect sizes revealed that sites with the largest changes in editing levels between living and postmortem DLPFC typically exhibited high editing levels (>30%). This also confirmed that 14 out of the 27 living-biased recoding sites were part of a collection of well-known functional sites on excitatory, inhibitory, and G-coupled protein receptors (e.g., *CYFIP2*, *NOVA1*, *GABRA3*, *GRIA2*)[2]. Living-biased recoding sites were also more strongly evolutionary conserved (phastCons) and mapped to genes with higher probability of being loss-of-function intolerant (pLI) relative to postmortem-biased recoding sites ($p = 0.002$, $p = 0.002$, respectively) (Fig. 3G), underscoring their physiological relevance.

### Annotating dynamically regulated sites across cellular, developmental, and disease scales

We next sought to annotate the cellular context as well as the developmental and disease relevance of the differentially regulated sites between living and postmortem DLPFC. Given the scarcity of known cell-specific A-to-I sites in the brain and the technical limitations of quantifying cell-specific sites from snRNA-seq data (Fig. S8, see also

ref. 13), we leveraged an independent resource of deeply sequenced neuronal and non-neuronal nuclei isolated from ten biological replicates across five postmortem cortical regions[32]. These data were subjected to RNA editing calling methods described above. In doing so, we generated an expanded catalog of cell-specific RNA editing sites across the cortex (*see* Supplemental Note 1 for details, Fig. S9–11). Subsequently, these findings, together with previously generated collections of A-to-I sites cataloged as cell type-specific, temporally regulated across brain development as well as sites disrupted in neurological disorders, were used to annotate differentially edited sites between living and postmortem DLPFC (Supplemental Data 4).

Postmortem-biased sites were enriched for A-to-I sites cataloged as non-neuronal cell type-specific ($p = 8.9 \times 10^{-7}$), including oligodendrocytes ($p = 2.6 \times 10^{-9}$) (Fig. 4A, B). Conversely, living-biased sites were significantly enriched for sites cataloged as neuronal cell type-specific ($p = 0.005$), including GABAergic neurons ($p = 0.04$) (Fig. 4A, B). A more focused analysis dissected the cellular specificity of differentially edited RNA recoding sites between living and postmortem DLPFC (Fig. 4C). This re-affirmed that living-biased recoding sites were neuronal-specific and featured several well-known

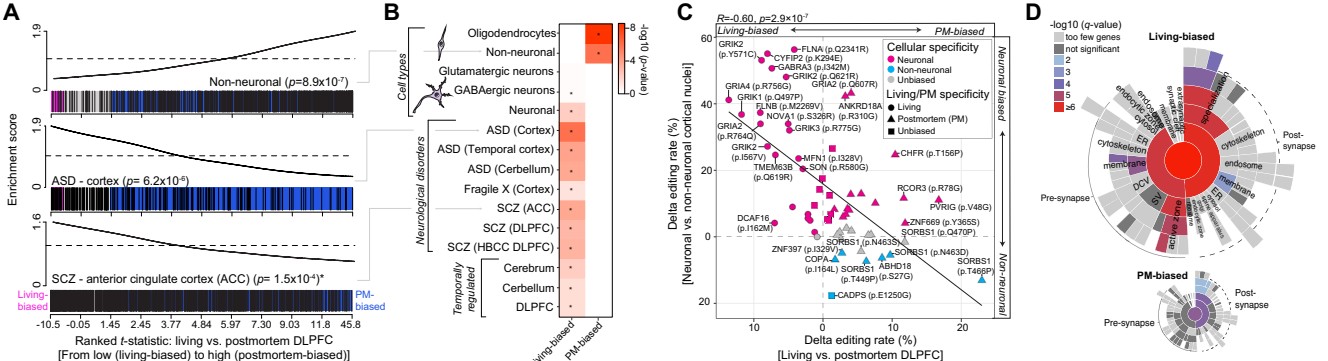

**Fig. 4 | Annotating dynamically regulated sites between living and postmortem DLPFC. A** CAMERA enrichment scores ($y$ axes) for three candidate sets of A-to-I editing sites along a ranked list of differentially edited sites ($t$ statistics; $x$ axis) between living and postmortem (PM) DLPFC, from highly living-biased (right; pink) to highly PM-biased (left; blue) ($x$ axes). Enrichment plots for non-neuronal sites (top, PM biased), sites disrupted in autism spectrum disorder (ASD) cortex (middle, living biased), and those disrupted in schizophrenia (SCZ) ACC (bottom, living biased). **B** Summary of all multiple test-corrected $p$ values ($-\log_{10}$) for all sets of RNA editing sites across cell types, neurodevelopmental disorders, and brain development. **A**, **B** CAMERA gene set enrichment $p$ values, quantifying the statistical significance of overrepresented A-to-I sites within the ranked living versus postmortem data. **C** Pearson's correlation and scatterplot of delta editing rates for cell-specific recoding sites ($y$ axis) versus delta editing rates for living/PM differences ($x$ axis). $Y$ axis description: Fluorescence activated nuclei sorted (FANS) neurons and non-neuronal cell populations were collected from 10 postmortem donors across five cortical regions (see Supplemental Note 1). **D** SynGO synaptic enrichment ($-\log_{10}$ $q$ value) for genes harboring living-biased editing sites (top) and genes harboring postmortem-biased sites (bottom). RNA-seq analysis encompassed 164 and 233 biologically independent samples from living and postmortem sources, respectively.

functional and highly conserved recoding sites, whereas postmortem-biased recoding sites were non-neuronal and mechanistically less understood. Living-biased sites were also significantly enriched for several additional functional categories, including enrichment for: (i) A-to-I sites disrupted in postmortem brain tissues from individuals with autism spectrum disorder (ASD) (cortex $p = 1.1 \times 10^{-6}$; temporal cortex $p = 1.0 \times 10^{-4}$; cerebellum $p = 1.5 \times 10^{-4}$) and schizophrenia (SCZ) (ACC $p = 1.5 \times 10^{-4}$; DLPFC $p = 7.6 \times 10^{-4}$; DLPFC HBCC $p = 4.8 \times 10^{-4}$); (ii) A-to-I sites with precise spatiotemporal regulation across human prenatal and postnatal brain development (DLPFC $p = 0.02$; cerebrum $p = 0.006$; cerebellum $p = 0.02$) (Fig. 4B); and (iii) living-biased sites preferentially mapped to genes enriched for postsynaptic organization and density, as well as presynaptic activity genes (Fig. 4D). Considering the high levels of editing in living tissues, these findings imply a connection between A-to-I editing sites biased towards living tissues and their functional importance and endorse the value of postmortem case/control studies to explore RNA editing in brain health and disease.

### Replicating postmortem-induced effects on A-to-I editing in independent transcriptomic resources

Altogether, 72,356 A-to-I sites were cataloged as either detected at different frequencies or altered in mean editing levels between postmortem and living DLPFC, here defined as 'LIV-PM sites'. To validate these findings, we first replicated the strong postmortem-bias in editing levels for these sites leveraging pseudo-bulk snRNA-seq data from a non-overlapping sample of 31 living and 21 postmortem DLPFC samples ($r = 0.56$) (Fig. S12). Next, we asked what fraction of the LIV-PM sites are commonly detected across four independent large-scale postmortem brain transcriptome consortia, which contain a diverse collection of anatomical regions, neurological disorders, and age ranges (GTEx project [$n = 1129$]; Mount Sinai Brain Bank [$n = 876$], PsychENCODE [$n = 251$]; BrainSpan [$n = 586$]). Collectively, these data have been extensively studied for their RNA editing properties and serve as cornerstone resources for A-to-I sites in the human brain[13,16,19,20,33] (Supplemental Data 3). Overall, we found a significant over-representation of LIV-PM sites routinely detected across these independent postmortem transcriptomic resources, whereby LIV-PM sites represented ~15–31% of all commonly detected A-to-I sites (Fig. S13). LIV-PM sites also exhibited stably high editing levels

(~40–46%), which were significantly higher (~7–20%) than all other detected A-to-I sites (Fig. S13), implicating a systematic postmortem-induced effect for these sites across diverse cohorts, anatomical regions, and ages.

### Profound increases of *Alu* editing in postmortem tissues explained by interferon activation and hypoxia

We next explored the biological processes that may best explain the profound postmortem biases in RNA editing. Gene set variation analysis (GSVA) computed single-sample scores for 10,493 Gene Ontology Biological Processes for each bulk tissue RNA-seq sample, which were regressed onto the AEI to identify biological processes predictive of alterations in global *Alu* editing (Fig. S14A). A total of 1688 biological processes were positive predictors of global *Alu* editing (FDR < 5%) and were broadly enriched for categories of innate immune and inflammatory responses, hypoxia, intracellular signaling, apoptosis, and cellular metabolism (Fig. 5A, B). Notably, biological processes that predicted the AEI were also strong predictors of living versus postmortem DLPFC ($r = 0.86$) (Fig. S14B, C). For example, the expression of genes that subserve the following biological processes stratified living from postmortem samples and were strongly predictive of changes in the AEI: 'inositol trisphosphate metabolism' (GO:0032957; $t$ statistic = 14.6, $q$ value = $1.2 \times 10^{-36}$), 'desensitization of G-protein coupled receptor (GPCR) signaling pathway' (GO:0002029; $t$ statistic = 12.1, $q$ value = $8.4 \times 10^{-27}$), 'regulation of cell cycle' (GO:0051726; $t$ statistic = 11.8, $q$ value = $1.5 \times 10^{-25}$), 'response to hypoxia' (GO:0001666; $t$ statistic = 9.3, $q$ value = $4.8 \times 10^{-17}$) and 'positive regulation of interferon-gamma (IFN-$\gamma$) signaling pathway' (GO:0060355; $t$ statistic = 8.7, $q$ value = $7.6 \times 10^{-16}$), among other innate immune responses (Fig. 5D). Furthermore, the gene expression profiles underlying these biological processes were also significantly elevated in postmortem relative to living DLPFC (Fig. 5C).

To functionally validate the observed associations, we quantified the AEI across two existing cellular and mechanistically related in vitro models[34,35]. First, we disentangled the influence of IFN-$\gamma$ signaling on global *Alu* editing in hiPSC-derived neural progenitor cells (NPCs) and mature neurons treated with IFN-$\gamma$[34]. Here, IFN-$\gamma$ significantly induced global *Alu* editing across acutely treated NPCs and chronically treated mature neurons relative to untreated conditions ($p = 0.0003$, $p = 0.02$, respectively) (Fig. 5E). Next, we quantified the effect of hypoxia on

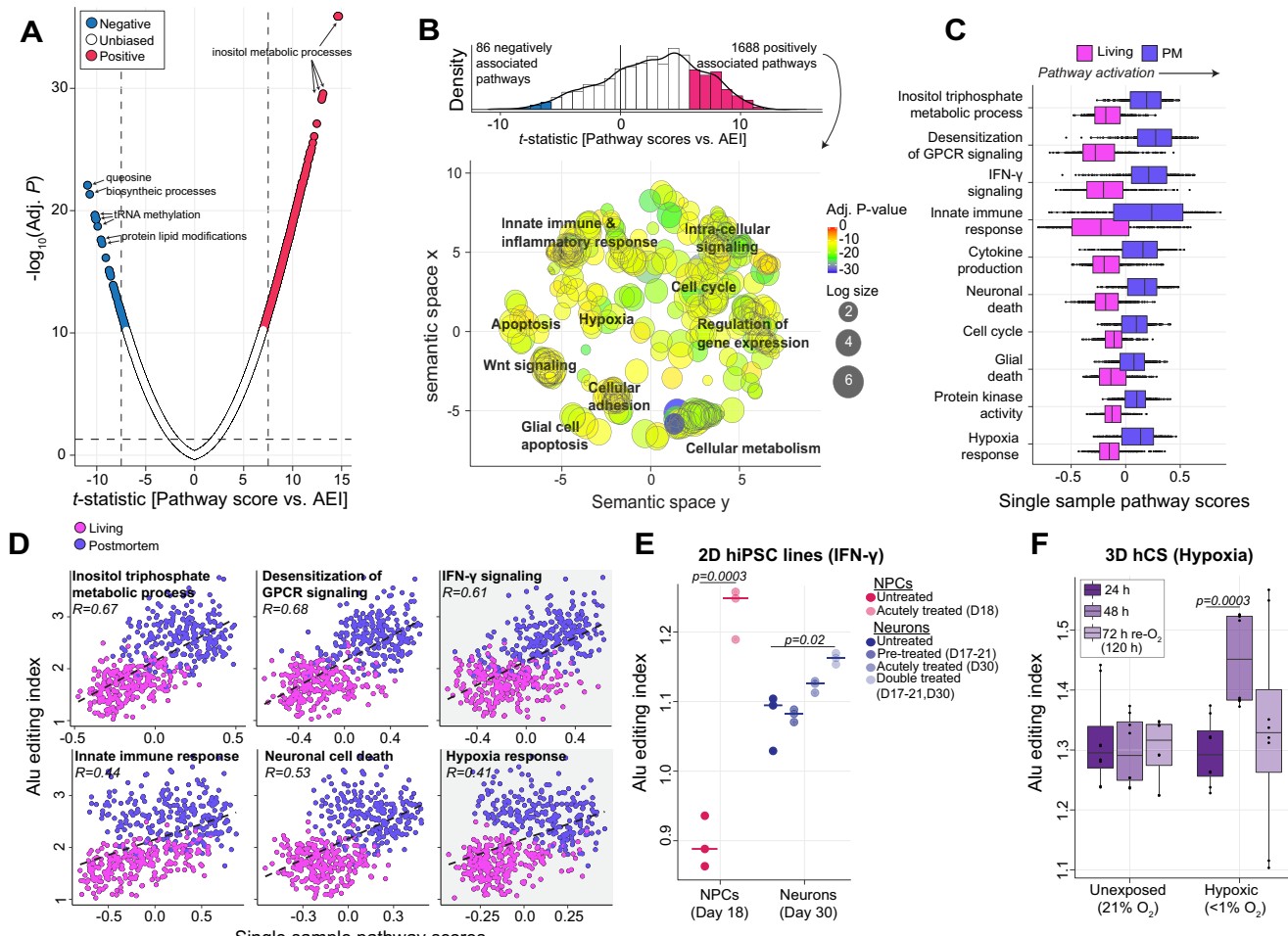

**Fig. 5 | Biological processes that explain differences in *Alu* editing. A** Single-sample scores were generated for 10,493 Gene Ontology Biological Processes. These pathway scores were regressed onto the AEI while covarying for known biological and technical factors. Reported BH adjusted *p* values and *t* statistics were derived from a moderated *t* test comparing pathway scores between living and postmortem tissue. The *t* statistics (x-axis) for each biological process relative to strength of association with the AEI (*y* axis; −log₁₀ adjusted *p* value). Pathways with an absolute *t* statistic >7 and FDR adjusted *p* value < 0.05 were deemed significant (blue, negative association; red, positive association). **B** A density distribution of *t* statistics illustrates most pathways are positive predictors of the AEI (top) and REVIGO semantic similarity representation of the top positive 1688 pathways (bottom,). Multiple broad groupings emerge that map to intra-cellular signaling, apoptosis, hypoxia, cellular metabolism, and innate immune/inflammatory responses. Colors indicate the adjusted *P* value of the enriched GO terms. The size of each bubble shows the GO terms with more significant *P* values. **C** Single-sample scores that represent top pathways from each cluster also predict differences between living and postmortem (PM) tissues. **D** Pearson's correlation coefficient illustrates regressions of single-sample pathway score onto the AEI for the top six candidate pathways. All Living Brain Project data encompassed 164 and 233 biologically independent samples from living and postmortem sources, respectively. **E** Validating the interaction between interferon-γ and the AEI. Two-dimensional (2D) hiPSC-derived neural progenitor cells (NPCs; day 18) and mature neurons (day 30) treated with interferon-gamma (IFN-γ) (PMID: 32875100). Data was generated by bulk RNA-seq from *n* = 3 biological replicates. **F** Validating the interaction between hypoxia and the AEI. Three-dimensional (3D) model of human cortical spheroids (hCS) exposed to differing degrees of hypoxia (PMID: 31061540). Data was generated by bulk RNA-seq from *n* = 8 biological replicates. **E**, **F** Two-sided linear regression was used to test for significance. All boxplots in this figure show the medians (horizontal lines), upper and lower quartiles (inner box edges), and 1.5× the interquartile range (whiskers).

---

global *Alu* editing in human cortical spheroids (hCS) exposed to hypoxic conditions (<1% O₂) and after 72 hours of reoxygenation versus unexposed (21% O₂)[35]. Hypoxic conditions significantly increased global *Alu* editing from 24 to 48 hours (*p* = 0.0003) and returned to baseline levels after 72 hours of reoxygenation (Fig. 5F). Notably, both models revealed significant increases in *ADAR* expression profiles (Fig. S15). These in vitro results support a mechanistic model, whereby IFN-γ and hypoxia induce global *Alu* editing in human neuronal models, perhaps explaining some of the observed increases in the AEI in postmortem DLPFC.

### Mapping context-dependent edQTLs between living and postmortem DLPFC

We next sought to elucidate RNA editing quantitative trait loci (edQTLs). Paired WGS data were used to detect SNPs that could influence A-to-I editing levels for 155 living and 195 postmortem DLPFC samples. Two cis-edQTL analyses were performed: (**1**) A *primary* cis-edQTL analysis fit A-to-I editing levels to SNPs while covarying for differences between living and postmortem tissues, sex, estimated neuronal content, RNA-seq QC metrics as well as eleven surrogate variables (PEER factors); and (**2**) An *interaction* cis-edQTL analysis tested for context-dependent effects between living and postmortem tissues (see Materials and Methods). A 1 Mb window (±) was defined to search for SNP-editing pairs of an editing site and identified a total of 4858 and 2362 unique editing sites with cis-edQTLs (eSites) from the primary and interaction analyses at FDR < 5%, respectively (Supplemental Data 5). eSites were largely non-overlapping between the two analyses and together comprised a total of 6895 unique editing sites. Each lead SNP was located close to their associated editing site (±200 kb) (Fig. 6A). eSites from the primary and interaction analyses

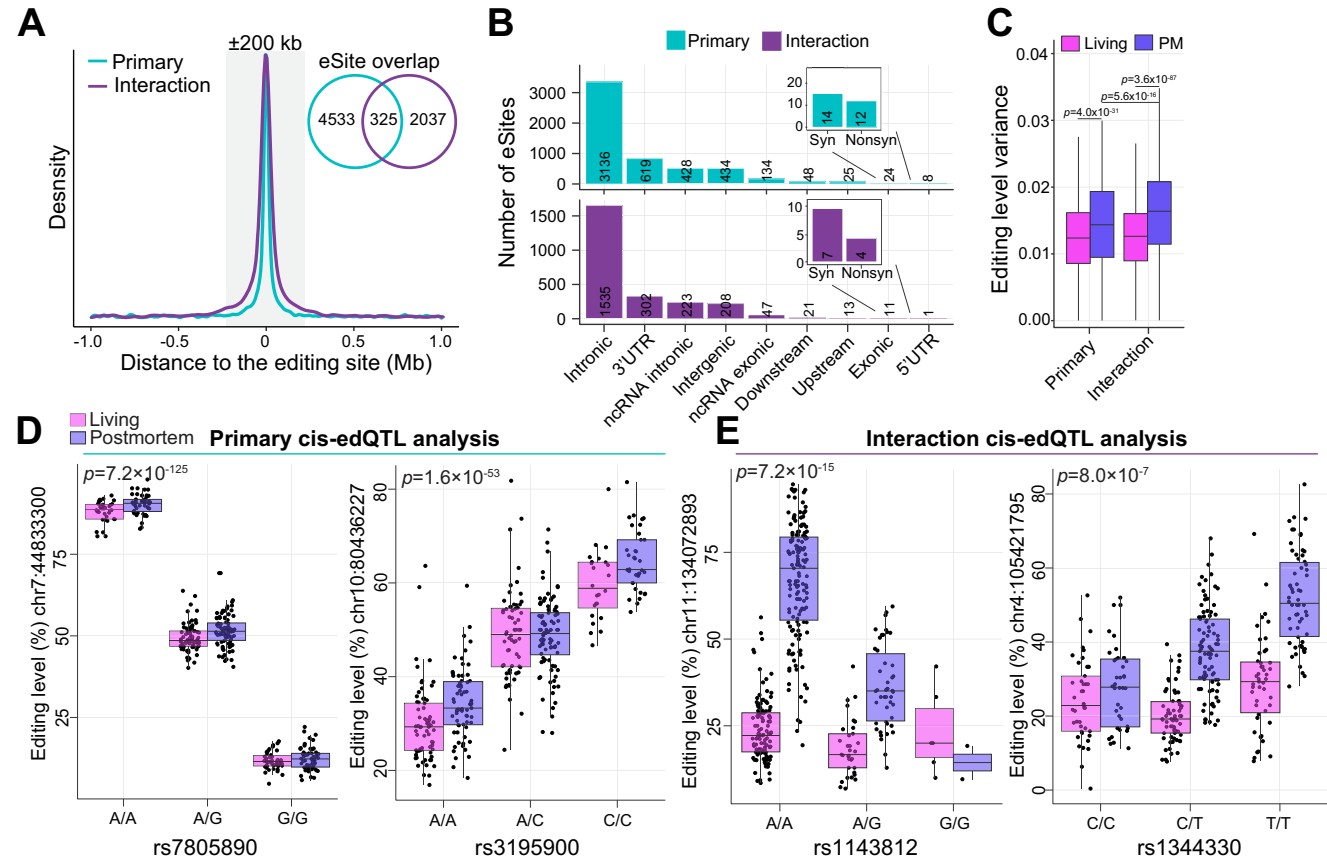

**Fig. 6 | Context-dependent cis-edQTLs between living and postmortem DLPFC.** **A** Distribution of the cis-edQTL associations evaluating the distance between eSites and common variants. The gray box indicates ±200 kb relative to the editing site. Inset Venn diagram depicts the overlap of eSites between the primary and interaction analyses. **B** eSite discovery (*y* axis) according to genic regions (*x* axis) for primary (top) and interaction (bottom) cis-edQTL analyses. **C** Editing level variance within living and postmortem DLPFC parsed by sites with primary and interaction cis-edQTLs. **D** Two examples of primary cis-edQTLs, in which editing levels (*y* axes)

are associated with common genotypes (*x* axes). **E** Two examples of interaction cis-edQTLs, in which different common genotypes (*x* axis) are associated with differing editing levels between living and postmortem DLPFC (*y* axes). **C**–**E** Two-sided linear regression was used to test for significance. Analyses encompassed 164 and 233 biologically independent samples from living and postmortem sources, respectively. All boxplots in this figure show the medians (horizontal lines), upper and lower quartiles (inner box edges), and 1.5× the interquartile range (whiskers).

mapped to introns and 3′UTRs with few in protein-coding sequences (Fig. 6B). Furthermore, editing levels for eSites, especially those from the interaction analysis, displayed higher variability in postmortem relative to living DLPFC (Fig. 6C). Most cis-edQTLs from the primary analysis explained a large percentage of editing level variability per site, explaining up to 90% editing level differences for a given site (Fig. 6D). Cis-edQTLs from the interaction analysis often had smaller effects, with ~83% illustrating postmortem-specific effects (Fig. 6E). Notably, ~2% eSites identified from the interaction analyses were commonly reported across previous cis-edQTL investigations in the postmortem brain[13,33,36] (Fig. S16), indicating that postmortem-related effects do not widely contribute to our current view of edQTLs in the brain.

## Discussion

The investigation of A-to-I editing and its biological significance in the mammalian brain has been restricted to the analysis of postmortem tissues. However, increasing evidence indicating molecular changes in response to ischemic exposures in the brain highlights the need for a more accurate understanding. Utilizing fresh brain tissue from living human donors provides an opportunity to investigate the brain without the confounds inherent to postmortem tissue analysis. Here, we present the first systematic study of A-to-I editing differences between postmortem and living human DLPFC. In doing so, we reveal more nuanced and accurate insights into the prevalence and multifarious

roles of A-to-I editing in the human brain, specifically: (1) elevated *ADAR* and *ADARB1* as well as widespread features of A-to-I editing are enriched in postmortem relative to living DLPFC; (2) these postmortem-related increases are pronounced in non-neuronal cell populations; (3) these changes also scale with elevated expression of gene that subserve biological pathways and molecular responses to human death, including IFN-γ signaling, hypoxia and intracellular metabolism; (4) systematic investigation of A-to-I sites highly edited in the living brain offers a unique and powerful framework to prioritize sites that are essential for brain function; (5) context-dependent cis-edQTLs reveal genetic variants with differing effects on A-to-I editing levels between postmortem and living DLPFC; and (6) despite widespread differences, well-powered cohorts of postmortem tissues remain a valuable resource for studying A-to-I editing and its roles in brain development and disease pathology. We discuss these points in turn below.

Mounting evidence of molecular responses to ischemic insults in the brain, coupled with the precise control of A-to-I editing and the crucial function of ADAR1 in regulating innate immunity, strongly implicate dysregulated A-to-I editing in postmortem brain tissues. Indeed, our analyses revealed a significant elevation of the AEI as well as *ADAR* and *ADARB1* expression in postmortem compared to living DLPFC (Fig. 2), which was confirmed on a per-cell basis (Fig. S5). To reduce the likelihood of confounding factors in the current study, our postmortem DLPFC cohort is matched to the extent possible to the

living cohort for key demographic and clinical variables (see full cohort description in Supplemental Data 1). Subsequently, the contribution of any additional factors on A-to-I editing variability were minimal in the current study yet warrants further discussion. For example, although A-to-I editing is enriched in neuronal cell types[13,14] and elevated proportions of neurons were predicted in postmortem DLPFC, differences in cellular frequencies had an overall small effect on A-to-I editing differences between postmortem and living DLPFC (<0.5%). In fact, microglia showed the most pronounced postmortem-induced effect on A-to-I editing (Fig. 2G), and these cells play key roles in regulating the brain's response to inflammation[37,38]. Microglia also express high levels of *ADAR*, which is interferon inducible[6–8,39] and likely explains why microglia are highly susceptible to A-to-I perturbations following human death. The length of PMI also had little influence on A-to-I editing increases (see also Fig. S3), which suggests that the impact of death on A-to-I editing is immediate, with subtle and non-systematic alterations occurring in the hours following death. This is also consistent with independent studies evaluating the influence of PMI on mRNA levels, whereby relatively few genes show significant changes in expression over extended PMI, and genes that do change do not follow an orderly pattern of expression[27,40]. Further quantification of RNA editing metrics throughout a molecular degradation assay of the human DLPFC[31] did not suggest that A-to-I editing is greatly impacted by tissue degradation (Fig. S4) nor strongly influenced by RIN (Fig. 2C). While postmortem tissues may indeed be confounded with elevated RNA degradation, the observed changes in the AEI and ADAR expression between living and postmortem tissues is likely not fully explained by such molecular factors. Future work to dissect such impacts is warranted. Finally, while disrupted A-to-I editing levels have been linked to a clinical diagnosis of PD[41], PD status had no effect on A-to-I editing variance between postmortem and living DLPFC (Figs. S2A and S14E). However, a deeper investigation into the influence of discrete neuropathological hallmarks on A-to-I editing profiles within living tissues is warranted and should be the focus of future studies.

Altogether, 72,346 A-to-I sites were cataloged as either preferentially enriched or differentially edited between postmortem and living DLPFC, with a significant postmortem-bias (Fig. 3). Replication of the LIV-PM sites across four large-scale postmortem brain transcriptome datasets supports the notion that postmortem-induced activation of *ADAR*- and *ADARB1*-mediated A-to-I editing occurs in a profound and systematic manner across diverse anatomical regions and data sets. In asking which biological processes might be driving the postmortem enrichment of A-to-I editing, we found that increased expression of genes subserving IFN-γ signaling, hypoxia, cellular metabolism and apoptosis in postmortem DLPFC were strongly correlated with an elevated AEI (Fig. 5). Independent studies analyzing alterations in gene expression after organismal death revealed an upregulation of similar pathways[27,40]. While not all these biological processes may be causal for increased editing, we validated the influence of IFN-γ signaling and hypoxia on global *Alu* editing levels using already existing RNA-seq data from human neuronal model systems (Fig. 5). Indeed, IFN-γ signaling is known to induce *ADAR*, leading to elevated RNA editing[6–8,39]. ADAR1 has also been shown to promote accumulation of HIF1A following oxygen depletion[42,43], supporting dynamic changes of A-to-I editing after hypoxic and re-oxygenated exposures. Furthermore, increased *Alu* editing has been reported across non-CNS human tissues collected from postmortem donors compared to those collected from donors while on mechanical ventilation[29], suggesting that hypoxic exposures have direct effects on RNA editing levels. Moreover, genes associated with inositol triphosphate metabolic processes were higher expressed in postmortem and appeared to be a positive predictor for A-to-I editing. Interestingly, inositol hexakiphosphate has been found to be in the core structure of ADAR2 and is crucial for ADAR2 activity[44]. More access to inositol triphosphate could therefore boost ADAR2 activity and lead to

elevated A-to-I editing in postmortem tissues. Overall, our results support a mechanistic model whereby postmortem-related mechanisms, including inflammatory and hypoxic responses, induce *ADAR* expression. In turn, this leads to an abundance of A-to-I editing commonly observed in postmortem brain tissue.

We also identified hundreds of A-to-I editing sites that were more highly edited in living brain tissues than in postmortem samples. These sites are enriched in neuronal synapses and are typically evolutionarily conserved, suggesting their functional relevance in brain activity (Figs. 3F, 4D). Notably, well-characterized neuronal recoding sites with unique functional properties were predominantly edited in living DLPFC, signifying their potential involvement in active neuronal processes such as synaptic plasticity. However, several living-biased recoding sites that are less functionally understood were also found and their contribution to brain function remains an open question for future research. For example, recoding site p.M2269V in *FLNB* exhibited neuronal-specificity and is known to be tightly regulated across stages of prenatal and postnatal human brain development[16]. In mouse brain, editing of Flnb is driven by ADAR2 and leads to less efficient splicing of the transcript[45]. While Flnb is highly edited outside the CNS, including in musculoskeletal tissues, uncovering the functional consequences of Flnb editing will be the next experimental challenge. Similarly, recoding site p.I328V in *MFN1* was neuronal-specific, and MFN1 is known to mediate mitochondria fusion process, and reduced editing on this transcript has been implicated in Alzheimer's disease[22]. Additionally, editing variations at the GRIA2 Q/R and R/G sites between living and postmortem DLPFC highlight how postmortem conditions may differentially influence the editing machinery, potentially releasing it from regulatory constraints or preserving certain functional editing activities related to neuronal maintenance, respectively. The discrepancy between the two sites might be attributed to their distinct functional roles and the different susceptibilities to postmortem changes.

Another important observation is that living-biased sites are enriched for A-to-I sites that exhibit tight spatiotemporal regulation throughout brain development[16] and are associated with neurological disorders when disrupted[19,20] (Fig. 4B). Therefore, our findings support the continuing value of postmortem brain tissues for RNA editing research in the context of disease pathobiology, as they preserve the integrity of editing sites implicated in brain function and pathology. However, while our results reinforce the relevance of postmortem case/control studies in reflecting the physiological state of RNA editing, they also highlight the necessity for discerning potential postmortem artifacts.

Integration of paired WGS with A-to-I editing levels in the form of cis-edQTLs provides unique opportunities to dissect how genetic variation regulates editing levels (Fig. 6). edQTLs explained up to ~90% editing level differences, with the majority of cis-edQTLs located in 3′-UTRs, which is consistent with previous reports by us and others[13,16,33,36]. To this end, a mechanistic model has been proposed whereby 3′-UTR bound miRNAs can alter gene expression levels from an edQTL locus via miRNA-mediated transcript degradation[33]. We also cataloged 2362 context-dependent cis-edQTLs across 1247 unique loci that differed between postmortem and living DLPFC. Editing levels for these sites were generally homogenous within living tissues with a significant edQTL in postmortem DLPFC, implicating these eSites are highly sensitive to ischemic insults. Importantly, these context-dependent cis-edQTLs accounted for a small fraction (<2%) of all cis-edQTLs currently documented in the postmortem brain literature[13,33,36], indicating that cis-edQTLs with differing postmortem and living effect sizes do not significantly skew the interpretation of current postmortem edQTL findings, supporting the utility of postmortem brain tissues for edQTL discovery. However, to further dissect such mechanisms, it will be critical for future work to greatly increase sample sizes of fresh biopsies.

Our study also presents some limitations, which warrant further discussion. First, living samples were collected using a novel sampling procedure for study participants undergoing DBS[30]. While we refer to these fresh biospecimens as living tissue, they may present their own caveats, including the possible influence of anesthesia and surgical procedures, which may induce injury and early immediate epitranscriptomic responses. Second, sampling living brain tissue comes with spatial and regional constraints. While living tissue sampling could be biased towards increased white matter collection and elevated proportions of non-neuronal cell types, variation in cell type composition had an insignificant effect on A-to-I editing variability, as discussed above. Third, study participants were largely over 60 years of age with a diagnosis of Parkinson's disease (PD) requiring neurosurgical intervention. Nevertheless, ~30% of the living DLPFC cohort does not have PD, and a clinical diagnosis of PD did not alter *Alu* editing. Moreover, both PD and control subjects were included in the postmortem cohort to match living and postmortem DLPFC according to clinical, demographic, and technical factors to minimize its potential confounding effect. Fourth, we cannot fully rule out the possibility that both living and postmortem tissues may have differing medication effects. Still, it is worth emphasizing that even in tightly controlled experiments conducted by us and others, the overall influence of low and high doses of antipsychotics and small molecules on A-to-I editing profiles in the brain is insignificant[19,46]. Finally, we note that cell type-specific A-to-I sites were cataloged from postmortem tissues[13,32], thus caution is warranted when interpreting their editing levels, especially for non-neuronal A-to-I sites, which are more vulnerable to postmortem-induced mechanisms.

In sum, we provide a large-scale systematic investigation of A-to-I editing in the living human brain, in which we propose a model whereby early immediate biological responses to human death, including activation of IFN-γ signaling and hypoxia, up-regulate the expression of *ADAR* and *ADARB1*. This is followed by a coordinated increase in transcriptome-wide A-to-I editing. Moreover, these profound A-to-I editing increases in postmortem tissues are distinct and independent from changes that may be related to any tested confounding effects. Further, investigation of A-to-I sites that are highly edited in the living brain offers a powerful framework to identify sites that are putatively functionally relevant for brain function. Critically, our findings do not negate but instead, provide missing context for using postmortem brain tissues in researching A-to-I regulation in brain health and disease. As we advance, the detailed molecular analysis of living brain tissues presents considerable challenges for large-scale study, yet it holds the potential to yield promising insights into the biology of RNA and A-to-I editing, which could transform our understanding of both the healthy and diseased human brain.

## Methods

### Ethics declaration
All human subjects research was conducted in accordance with the criteria set by the Declaration of Helsinki, carried out under STUDY-13-00415 of the Human Research Protection Program at the Icahn School of Medicine at Mount Sinai and approved by the Icahn School of Medicine at Mount Sinai's Institutional Review Board. The study design and conduct complied with all relevant regulations. Research Participants in the living cohort provided informed consent for sample collection, genomic profiling, clinical data extraction from medical records, and public sharing of de-identified data.

### Experimental model and subject details
**The Living Brain Project.** The current study is anchored around *state-of-the-art* LBP data comprised of multi-omic paired WGS, bulk-tissue RNA sequencing (RNA-seq), and single nuclei RNA-sequencing (snRNA-seq) of living and postmortem DLPFC samples (see full cohort description in Supplemental Data 1). All the data from living and postmortem samples studied in the current report have been introduced in detail in complementary LBP reports: bulk RNA-seq was introduced in Liharska et al.[30], snRNA-seq was introduced in Vornholt et al.[47], and the WGS data is introduced in Kopell et al.[48]. Each of these datasets is more fully described in each respective body of work. We briefly describe these protocols here:

Regarding ascertainment of living samples, fresh tissues were obtained during a DBS electrode implantation procedure at the Icahn School of Medicine at Mount Sinai. For the procedure, a burr hole was created in the frontal bone to access the cortical surface. A unique modification was made to the DBS procedure, which allowed collection of a small DLPFC biopsy for the LBP, as described in Liharska et al.[30]. These biopsies were immediately preserved in RNAlater or on dry ice and stored at −80 °C. Most living samples were obtained from individuals with PD, with non-PD samples collected for other DBS indications. Unilateral and bilateral biopsies were obtained. Informed consent was obtained from donors or their next-of-kin, and diagnoses were based on medical records, questionnaires, and neuropathological evaluations.

Regarding ascertainment of postmortem tissues, postmortem DLPFC samples were obtained from three different brain banks, matching them with living samples in terms of age, sex, and clinical diagnosis to the extent possible. Three separate brain banks were utilized, specifically Harvard Brain and Tissue Resource Center, the New York Brain Bank and Columbia University, and the University of Miami Brain Endowment Bank. Standard protocols were followed for processing postmortem samples, including dissection, freezing, and storage. Donors provided consent before death, and diagnoses were made based on various sources of information, including medical records and neuropathological evaluation. The New York Brain Bank at Columbia University specifically focused on donors with age-related neurodegenerative diseases and those without neurological or psychiatric impairments.

In the current study, we leveraged DLPFC bulk tissue RNA-seq data from a total of 251 living DLPFC samples (comprising 164 biological replicates [66 male and 98 female]) together with 233 postmortem DLPFC samples (comprising 233 biological replicates [148 male and 85 female]). A non-overlapping sample of 31 living DLPFC tissues was subjected to snRNA-seq (comprising 22 biological replicates [18 male and 4 female]) together with 21 postmortem DLPFC samples (comprising 21 biological replicates [13 male and 8 female]). Finally, WGS was used from 155 living and 195 postmortem DLPFC samples with paired bulk tissue RNA-sequencing data. All data are available on Synapse: Syn26337520.

**In vitro validation models.** Two different in vitro models were studied. The first study consisted of 18 RNA-sequencing samples generated from hiPSC-derived neural progenitor cells (day 18) and mature neurons (day 30) acutely or chronically treated with IFN-γ or left untreated[34]. All data are available on Synapse Syn18934100. The second study consisted of 48 RNA-sequencing samples generated from hCS exposed to hypoxic conditions (<1% $O_2$; at 24 hours and 48 hours) and after 72 hours of reoxygenation versus unexposed (21% $O_2$)[35]. All data are available at GEO GSE112137.

**Genotype tissue expression (GTEx) project.** Approved access to the GTEx Project was obtained through the database of Genotypes and Phenotypes (dbGaP) phs000424.v8 [https://www.ncbi.nlm.nih.gov/projects/gap/cgi-bin/study.cgi?study_id=phs000424.v8.p2] (Supplemental Data 1). A total of 1390 raw fastq files were analyzed, comprising bulk tissue transcriptomes across thirteen different postmortem brain regions (see Supplemental Data 1 and 3). The total number of biological replicates per region is as follows: Anterior Cingulate Cortex (ACC), $n = 95$; Amygdala, $n = 80$; Basal ganglia, $n = 133$; Cerebellar hemisphere, $n = 118$; Cerebellum, $n = 145$; Cortex, $n = 125$;

Frontal cortex, $n = 114$; Hippocampus, $n = 107$; Hypothalamus, $n = 99$; Nucleus accumbens, $n = 124$; Putamen, $n = 104$; Spinal cord, $n = 73$; Substantia nigra, $n = 73$.

**Mount Sinai Brain Bank (MSBB).** A total of 876 bulk tissue RNA-sequencing samples across four different postmortem cortical areas were obtained from the MSBB through the AMP-AD Synapse Web Portal[49] (Supplemental Data 1). The total number of biological replicates per region is as follows: Brodmann area (BM) 10, $n = 253$; BM 22, $n = 218$; BM 44, $n = 218$; BM 36, $n = 187$. These data comprised postmortem tissues from individuals with various stages of neurodegeneration. All data are available on Synapse Syn7416949 [https://www.synapse.org/#!Synapse:syn7416949].

**PsychENCODE.** A total of 251 bulk tissue RNA-sequencing samples across three postmortem anatomical regions were investigated (vermis, temporal cortex (BA41 42), and frontal cortex (BA9))[20] (Supplemental Data 1). These data comprised postmortem tissues from individuals with autism spectrum disorder as well as a matched control group. All raw FASTQ files were downloaded from Synapse under accession number Syn8365527.

**BrainSpan.** A total of 585 bulk tissue RNA-sequencing samples covering prenatal and postnatal development periods (8 post-conception weeks until 40 years of age) and 16 postmortem anatomical regions were included in the current study[50] (Supplemental Data 1). These developmental brain samples comprised of both cortical and subcortical structures, including the amygdala (AMY), primary auditory cortex (A1C), cerebellar cortex (CBC), dorsolateral prefrontal cortex (DFC), hippocampus (HIP), posterior inferior parietal cortex (IPC), inferior temporal cortex (ITC), primary motor cortex (M1C), mediodorsal nucleus of thalamus (MD), medial frontal cortex (MFC), orbital frontal cortex (OFC), primary somatosensory cortex (S1C), posterior superior temporal cortex (STC), striatum (STR), primary visual cortex (V1C), and ventrolateral frontal cortex (VFC). Raw FASTQ files were downloaded from Synapse under accession number Syn8298777 [https://www.synapse.org/#!Synapse:syn8298777].

**FANS-derived neuronal and non-neuronal nuclei.** A total of 100 RNA-sequencing data were leveraged from neuronal and non-neuronal nuclei isolated via FANS[32] (Supplemental Data 4). NeuN (also known as RNA-binding protein RBFOX3) is a well-established marker of neuronal nuclei and was used to isolate neuronal (NeuN+) from non-neuronal nuclei (NeuN−). NeuN+ and NeuN− nuclei were sampled across five postmortem brain regions from ten biological replicates (BM 10, 17, 22, 36, 44). All data are available on Synapse Syn25716684 [https://www.synapse.org/#!Synapse:syn25716684/wiki/610496].

**Molecular degradation assay of the human DLPFC.** A total of 20 RNA-sequencing samples were downloaded from an existing study examining the molecular degradation of the human DLPFC[31] (NCBI BioProject Number: PRJNA389171 [https://www.ncbi.nlm.nih.gov/sra?linkname=bioproject_sra_all&from_uid=389171]). Postmortem DLPFC tissue was left at room temperature (off ice) at a series of subsequent time-points (0, 15, 30, and 60 minutes) followed by RNA extraction, RiboZero RNA-seq library preparation and sequencing. We used these transcriptome samples because the approach for RNA-seq library preparation is similar to the methods reported in the current study.

## Sample processing, quantification, and statistical analyses
### LBP sample batch assignment, RNA processing, and RNA-sequencing.
To minimize batch effects, all living and postmortem samples were processed together for RNA sequencing at the Icahn School of Medicine in New York City. First, a randomization algorithm was employed to create batches for RNA extraction, cDNA library preparation, and RNA sequencing. The algorithm aimed to minimize correlations between batch assignments and living and postmortem status, ensuring an even distribution of samples across processing steps. Next, approximately 5-10 milligrams of each LIV and PM sample were used for RNA extraction. A reference piece of postmortem brain tissue was used to standardize the aliquoting process, eliminating the need to weigh individual samples and prevent thawing. To maintain a uniform temperature, a cryostat set at −20 °C was used, and all equipment was treated with RNase Zap to prevent RNA degradation. After aliquoting, samples were homogenized in TRIzol, and RNA extraction was primarily carried out with the RNeasy Kit. RNA integrity was assessed, and only specimens with a RIN greater than 4.0 were sent for sequencing. Following, CDA libraries and RNA sequencing were performed at Sema4 in Stamford, CT. Libraries were prepared using the TruSeq Stranded Total RNA with Ribo-Zero Globin Kit, and sequencing was conducted on the NovaSeq 6000 System. Two S4 flow cells were used for each sequencing batch to achieve the targeted depth of 100 million paired-end reads per sample library. Sequencing was carried out in two waves, with the first wave reaching a depth of 50 million paired-end reads and the second wave, over a year later, sequencing the remaining samples to a depth of 100 million, with an additional 50 million reads generated for samples from the first wave. Further detailed materials and methods regarding sample collection and data generation for LBP samples are outlined in Liharska et al.[30].

**RNA-sequencing data and quality control.** All RNA-sequencing data were processed and mapped to the GRCh38 primary assembly with GENCODE gene annotations v30 using STAR (v.2.7.2a)[51]. Picard v2.22.3 tools marked duplicate reads and gathered RNA-sequencing short read metrics and distributions (https://github.com/broadinstitute/picard/). STAR produced a coordinated-sorted mapped BAM file for each sample, which was used to study features of RNA editing. For gene expression analyses, featureCounts (v1.6.3)[52] quantified gene expression, and unspecific filtering removed lowly expressed genes with less than 1 count per million in at least 10% of samples. The resulting raw counts were normalized with voomWithDreamWeights() from the variancePartition R package (v1.20.0)[53] and used for downstream analyses. Linear mixed models from the R package variancePartition were also used to characterize and identify biological and/or technical drivers that may affect the observed AEI. This approach quantifies the main sources of AEI variation attributable to differences in biological factors (e.g., clinical diagnoses, age, sex ethnicity) and technical factors (e.g., RIN, batch). For further details on bulk tissue RNA-sequencing data generation and pre-processing, we refer the reader to Liharska et al.[30].

**snRNA-seq data generation and quality control.** For isolating single cells from fresh brain tissue samples ($n = 31$), we utilized the Adult Brain Dissociation Kit (Miltenyi Biotech, #130107677), which is particularly suited for processing live tissue. This kit enables the enzymatic breakdown of the tissue while maintaining the integrity and viability of the cells, which is crucial for subsequent cell-specific analyses. The process included washing the tissue, enzymatic digestion, and mechanical dissociation to ensure a high yield of viable cells. After dissociation, the cells were further purified using filtration and centrifugation steps provided in the kit's protocol, including the removal of myelin and debris, to ensure a clean sample for downstream applications. For extracting nuclei from frozen brain samples ($n = 21$), we chose the Minute Single Nucleus Isolation Kit (Invent Biotechnologies, #BN-020), which is designed to handle the challenges posed by frozen tissue. This kit's protocol is optimized to protect and isolate nuclei, which are less susceptible to damage from the freeze-thaw cycle than whole cells. The kit facilitates the gentle extraction of nuclei while preserving RNA integrity, which is essential for accurate transcriptomic profiling from postmortem tissue. Subsequent gene

expression profiling for both isolated cells and nuclei was carried out using the Chromium Single Cell 3′ Gene Expression platform with Next GEM reagents (10X Genomics, #CG000204 Rev D), which allows for high-throughput analysis and precise barcode assignment to individual cells or nuclei. The integrity of the synthesized cDNA was verified through meticulous quality control checks. The libraries produced were then sequenced on the NovaSeq system using the NovaSeq 6000 S2 Reagent Kit (Illumina, #20028315), adhering to a sequencing protocol that ensures comprehensive coverage. For more details on snRNA-seq data generation, we refer the reader to Vornholt et al.[47].

For all snRNA-seq data, *CellRanger* software (v7.0) performed genome alignment using 3′ gene expression chemistry against the GRCh38 primary assembly, generated barcode/UMI counts, and cell filtering to create mapped BAM files and feature-barcode matrices. Reads mapped to introns were incorporated into final count matrices to include both pre-mRNA and mRNA, which is representative of nuclear RNA populations. SoupX[53,54] was applied to remove contaminating ambient RNA. To optimize cell classification and reduce unwanted variance, we quality-filtered, normalized, and scaled data according to Seurat's guidelines[55]. For these data, we used a set of previously implemented methods consisting of the following steps. First, a cell was excluded if the number of expressed genes was <200, with the number of UMI <200, or the percentage of mitochondria reads <1%. The normalization method was LogNormalize with a scale factor of 10,000. The linear regression was performed using the percentage of mitochondrial reads as a variable. Second, the FindVariableFeatures(selection.method= vst, nfeatures = 2000) function from Seurat R package (v4.2.0)[55] was used to identify highly variable gene features. Third, principal component analysis and uniform manifold approximation and projection (UMAP) performed unsupervised clustering of each cell type. Hierarchical clustering was manually checked along with the top-ranked genes in each cell cluster to determine cellular specificity based on well-known gene markers to verify the assignment of cell types and subtypes. Finally, cells with inconsistent assignments were pooled into their corresponding cell type cluster based on shared transcriptome-wide profiles using CellSelector() from Seurat R package (v4.2.0). Ultimately, we report on nine major cell-type clusters defined by canonical cell-type markers.

**Generating an AEI**. For RNA-seq data, the AEI method v1.0 computed the AEI[29] using a STAR-mapped BAM file as input. The AEI is computed as the ratio of edited reads (A-to-G mismatches) over the total coverage of adenosines in *Alu* elements and is a robust measure that retains the full *Alu* editing signal, including editing events residing in low-coverage regions with a low false discovery rate. The resulting metric is multiplied by 100, so the index describes the percentage level of editing. The predetermined genomic regions were set to all SINE/*Alu* repeats using the *Alu* bed table of the UCSC genome browser, where most A-to-I editing occurs in mice. Common genetic variation was also discarded using coordinates from UCSC genome browser (hg38 CommonGenomicSNPs150). Notably, we have applied these metrics to hundreds of independent samples, and this method has proven to be highly robust and scalable across postmortem brain RNA-seq samples from unique studies and library preparation protocols[16,29].

For snRNA-seq data, the AEI method v1.0 was applied in two different ways. First, the AEI was quantified, ignoring all cell barcoded information, thereby analyzing each mapped snRNA-seq BAM file as a pseudo-bulk tissue. Second, using the nine cell type annotations identified via unsupervised dimensionality reduction, cell-specific barcodes were used to parse each pseudo-bulk snRNA-seq BAM file into cell type-specific BAM file (i.e., pseudo-bulk pooling of cell types per donor). The AEI was then computed for each cell type-specific mapped BAM file for each donor.

**Cellular deconvolution of bulk tissue**. Cell type deconvolution of the bulk tissue RNA-sequencing data was performed on raw gene count data using dtangle[56] and a scRNA-seq cell type reference panel from the DLPFC[57], including GABAergic and glutamatergic neurons, oligodendrocytes, astrocytes, and microglia. The sum of GABAergic and glutamatergic neuronal cell type proportions was used as proxy of total neuronal fraction.

**RNA editing site detection and annotation**. A two-step approach was used to quantify high-quality A-to-I sites from sorted mapped bam files:

1. We first quantified A-to-I sites de novo to facilitate the discovery of A-to-I sites not yet cataloged in existing RNA editing databases. Here, JACUSA2[58] was applied with the following parameters: -p 10 −a D, M, Y, E, -m 20. All analyses considered read strandedness when appropriate.

2. Next, we applied a supervised approach to query nucleotide coordinates for A-to-I sites already cataloged through REDIportal[59], A-to-I sites cataloged across human brain cell types[13], and an extensive list of A-to-I recoding sites[60]. Here, the samtools mpileup function was used to query editing levels of known sites, as shown prior[13,16,19]. This secondary supervised approach was applied to ensure the identification and inclusion of well-known sites into downstream analyses.

Subsequently, filtering steps were applied to retain only high-quality, high-confident bona fide A-to-I sites[16]. Briefly, the following sites were removed: (i) multi-allelic events; (ii) sites mapping to homopolymeric regions or black-listed genomic regions in the genome[61]; (iii) sites mapping to common genomic variation in dbSNP(v150) and those in gnomAD with minor allele frequency >0.05; (iv) sites mapping to high confidence heterozygous or homozygous genomic calls using paired WGS data; (v) de novo called sites adjacent to read ends and splice sites; (vi) de novo called sites with coverage was below ten reads, edited read coverage was below three reads and an editing ratio below 1%; (vii) supervised sites with coverage below five reads and the number of edited reads below three. Following, the remaining sites were annotated using ANNOVAR[62] to gene symbols using RefGene, repeat regions using RepeatMasker v4.1.1, known RNA editing sites using the most recent version of REDIportal and conservation metrics were gathered using phastCons from the PHAST package[63].

**Detecting dynamically regulated A-to-I sites**. Two different approaches were used to identify dynamically regulated A-to-I sites. First, sites observed at significantly different population frequencies were computed using a two-proportions z-test via the prop.test() function in R. This result returns the value of Pearson's chi-squared test statistic, a *p* value, 95% confidence intervals, and an estimated probability of success. We required that all significant sites must have >20% difference in detection rates between living and postmortem DLPFC and no >10% detection levels in the comparison group for which the A-to-I site is depleted. Second, to identify sites with significantly different mean editing levels between living and postmortem DLPFC, linear modeling via the *limma* R package[64] was implemented and adjusted for the possible influence of the following covariates: RNA editing levels -Neurons + Sex + Age + RIN + 3′ read bias + percent mRNA bases + median insert sizes + strand balance + batch. The duplicateCorrelation() function was used to model donor (i.e. technical replicates) as a repeated measure. Additional models were fit covarying for the influence of *ADAR, ADARB1,* and/or *ADARB2*. All significance values were adjusted for multiple testing using the Benjamini-Hochberg (BH) method to control the false discovery rate (FDR). Sites passing a multiple test-corrected *p* value < 0.05 were labeled significant. This approach was applied to all

bulk RNA-seq and pseudo-bulk snRNA-seq data from the LBP and FANS-derived neuronal and non-neuronal nuclei.

**Quantifying A-to-I sites across large-scale postmortem brain consortia.** To query A-to-I sites from existing large-scale postmortem transcriptomic resources outlined above (GTEx, MSBB, PsychENCODE, BrainSpan), raw FASTQ files were processed and mapped to the GRCh38 primary assembly with GENCODE gene annotations v30 using STAR (v.2.7.2a)[51]. Picard v2.22.3 tools marked duplicate reads and gathered RNA-sequencing short read metrics and distributions. STAR produced a coordinated-sorted mapped BAM file for each sample, which was used to study features of RNA editing. Next, because these cohorts have been extensively studied for the RNA editing properties, we applied a supervised approach (samtools mpileup) to quantify millions of known sites from three main resources: REDIportal[59], A-to-I sites cataloged across human brain cell types[13] and an extensive list of A-to-I recoding sites[60]. The supervised also required that all A-to-I sites must have a coverage of at least 5 reads and 2 edited reads to be considered for downstream analyses.

**Quantifying A-to-I sites in postmortem neuronal and non-neuronal nuclei.** It is acknowledged that individual A-to-I site identification using snRNA-seq presents difficulties due to limitations such as low capture efficiency and sequencing depth, resulting in partial coverage of the genome[65]. Our prior research has discussed these technical challenges in depth, particularly in the context of human cortex snRNA-seq data[13]. Furthermore, snRNA-seq data are inclined to exhibit an overrepresentation of intronic editing due to the predominance of nuclear RNA, which contrasts with bulk tissue sequencing that includes both nuclear and cytoplasmic fractions[66–68]. To this end, we incorporated deep RNA-seq from florescence-activated nuclei sorted (FANS)-derived neuronal and non-neuronal cortical nuclei from a prior study[32], which were mapped to the GRCh38 primary assembly with GENCODE gene annotations v30 using STAR (v.2.7.2a)[51]. This generated a coordinated-sorted mapped BAM file for downstream analyses. Next, we applied both a de novo caller (JACUSA2) together with a supervised approach (samtools mpileup), as described above, for all the LBP bulk RNA-sequencing data. Because of the deep level of sequencing performed on these cell populations, we applied the de novo caller to ensure the capture of cell-type-specific A-to-I sites that are not already cataloged in existing databases. All subsequent filters and thresholds, as described for the LBP bulk RNA-sequencing data, were also applied here.

**Enrichment for cellular, developmental, and disease-related sites.** Correlation-adjusted mean rank (CAMERA) gene set enrichment[69] was performed using the resulting sets of differential editing summary statistics between living and postmortem tissues. Here, we used CAMERA to perform a competitive editing set (i.e., a set of curated A-to-I sites) rank test to assess whether the sites in each editing set were highly ranked in terms of differential editing relative to sites that are not in the editing set. For example, CAMERA first ranks editing level differences in living cortical tissues relative to postmortem cortical tissues. Next, CAMERA tests whether the user-defined editing sets are over-represented toward the extreme ends of this ranked list. After adjusting the variance of the resulting editing set test statistic by a variance inflation factor that depends on the site-wise correlation (which we set to default parameters, 0.01) and the size of the set, a p value is returned and adjusted for multiple testing. We used this function to test for enrichment of editing sets derived from three major brain cell types[13], human brain development[16], SCZ[19], Fragile X Syndrome, and ASD[20].

**Single-sample pathway activation scores.** The GSVA R package[70] was applied to VOOM normalized gene expression data to generate gene set-centric activation scores for each transcriptome sample, converting a matrix of genes to gene sets. Gene set activation scores were generated across a well-curated list of 10,493 Gene Ontology (GO) Biological Processes. Subsequently, linear modeling via the *limma* R package[65] tested single-sample pathway activation scores for associations with the AEI, as well as for differences in these scores between living and postmortem samples, while adjusting both analyses for the covariates described above. Gene sets passing a multiple test-corrected p value < 0.05 with an absolute t statistic >7 were labeled significant. Gene sets annotated as positive and negative predictors of global *Alu* editing were subjected to REVIGO semantic similarity[71] to reveal consensus groups of gene sets with similar gene content.

**Identification of RNA editing quantitative trait loci.** Cis-edQTLs were identified for all high-quality common variants within 1 Mb (±) of an editing site using the fastQTL permutation-based analysis[72]. Two different analyses were run: (1) A primary edQTL analysis testing relationships between editing levels and SNPs while covarying for differences between living and postmortem tissues, sex, estimated neuronal content, the top eleven PEER factors, and a series of RNA-sequencing metrics (median 3′ bias, percent mRNA bases, batch, median insert size, strand balance). This analysis was run using a total of 10,000 permutations: (2) An interaction analysis testing context-dependent effects between living and postmortem tissues while adjusting for the covariates above. This analysis was run using a total of 1000 permutations. For the results of each analysis, all SNP-variant pairs with p value < 0.05 obtained via beta approximation were deemed significant and used for downstream analyses.

**Reporting summary**
Further information on research design is available in the Nature Portfolio Reporting Summary linked to this article.

## Data availability
The data supporting the findings of this study are available from the corresponding authors upon request. The LBP RNA-sequencing data generated in this study have been deposited in the Synapse database under accession code syn26337520. Secondary data analysis was carried out for several additional data resources, which are also publicly available with accession codes described above within each corresponding methods section. In brief, datasets subjected to secondary data analysis are available at Synapse under the accession numbers syn18934100, syn7416949, syn8365527, syn8298777, syn25716684, at GEO under the accession number GSE112137, and at dbGaP under the accession number phs000424.v8.

## Code availability
Code is available at GitHub (https://github.com/BreenMS/Living-Brain).

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

## Acknowledgements

The LBP study participants are commended for their important role in science. The following centers, programs, departments, and institutes within the Icahn School of Medicine at Mount Sinai supported the work: Charles Bronfman Institute of Personalized Medicine; Department of Neurosurgery; Department of Genetics and Genomic Sciences; Department of Psychiatry; Department of Neuroscience; Department of Medicine; Friedman Brain Institute; Seaver Autism Center for Research and Treatment; Center for Disease Neurogenomics. Postmortem samples from Harvard Brain Tissue Resource Center and the University of Miami Brain Endowment Brain were acquired under the National Institutes of Health NeuroBioBank request number 543. Postmortem samples from the New York Brain Bank of Columbia University were acquired under request number 1962. The following individuals are acknowledged for their advice and feedback over the course of the study: Elenita Sambat, Michael O. Hebb, Wayne Goodman, Michael Donovan, Olha Fedoryshyn, Mary Fowkes, Vahram Haroutunian, Milind Mahajan, Sabina Berretta, Etty Cortes, Jean-Paul Vonsattel, Eric J. Nestler, and Dennis S. Charney. This work is dedicated to Pamela Sklar. This work was supported by the NIA (1R01AG069976-01, Charney AW; 5R03AG080170-02; Breen, MS) and the Michael J. Fox Foundation (18232, Charney AW).

## Author contributions

Conceptualization: M.S.B., B.H.K., P.S., A.C. Sample collection, curation, and data management: B.H.K., A.C., J.F.F., P.D., E.C., K.Z., E.M., B.F., L.W., H.S., Li.L., Lo.L., B.S., V.C., P.K., C.F., J.J., M.K.R., J.S., G.N.N., B.Z., E.S., P.R. Data generation: N.D.B., E.P., S.Z., L.L., E.V., D.L. Formal analysis: M.R.S., N.D.B., A.B.G., G.G., R.C.T., Y.J.P., M.W., D.K., A.Y., E.P., S.Z., Lo.L., E.V., D.L., M.S.B. Writing – original draft: M.R.S., A.B.G., G.G., A.Y., P.A., M.S.B., A.C. Funding acquisition and Supervision: M.S.B. and A.C.

## Competing interests

Girish N. Nadkarni is a founder of Renalytix, Pensieve and is a consultant to Renalytix, Heart Test Laboratories and Pensieve. He serves as a scientific advisory board member for Renalytix, Heart Test Laboratories and Pensieve. He also has equity in Renalytix, Pensieve and Verici. He has also consulted for GSK, Reata, Cambridge Capital and GLG consulting. Brian H. Kopell consults and receives consulting fees from Medtronic, Abbott, and Turing Medical. Eric Schadt has equity in, receives a salary for, and is a Corporate Executive Leader for Pathos. He is also a Board of Directors member for Sage Bionetworks and a Board of Directors member for 4YouandMe, a non-profit institute.
