## [Peer Review File · Nature Communications]

Divergent landscapes of A-to-I editing in postmortem and living human brainEditorial Note: This manuscript has been previously reviewed at another journal that is not operating a transparent peer review scheme. This document only contains reviewer comments and rebuttal letters for versions considered at *Nature Communications*.

REVIEWERS' COMMENTS

Reviewer #1 (Remarks to the Author):

The authors have made significant improvements to their already impressive and important work, addressing all of the referee concerns. Therefore, I strongly recommend accepting the paper.

Reviewer #3 (Remarks to the Author):

The authors have addressed my major concerns with the additional analyses and extensive discussion.

Minor points

1. I think that many confusions about the tissues came from their wording: "matched" samples between living and postmortem tissues. However, as they admitted in the discussion, living tissues were biased toward PD patients, and so on. When applicable, how about using "partially matched"?
2. I appreciated that the authors made it clear that the degree of editing level differences between living and postmortem tissues were not insignificant, especially in non-neuronal cell types. I suggest they emphasize this point while appreciating the utility of post-mortem tissues in future studies. For example, this point can be added to the last sentence of the introduction.
3. Regarding Figure 3B, although the volcano plot provides editing level differences with statistical significance, it will be helpful for readers to see the MA plot to see the relation between average editing levels and editing level differences.
4. "A-to-G editing" has been used in the main text. It is advisable to use "A-to-I" for consistency.
5. Many p-values were reported as values only: At least, the authors should mention what statistical tests were used and the number of samples used.
6. In the 6th paragraph of the discussion: "We also catalogued 2,362 context-dependent cis-edQTLs across 1,247 unique that differed between postmortem and living DLFP." Does 1,247 indicate the number of loci?

Below we provide point-by-point responses, which fully address all minor points raised by Reviewer 3.

Reviewer #3 (Remarks to the Author):

The authors have addressed my major concerns with the additional analyses and extensive discussion.

Minor points:

1. I think that many confusions about the tissues came from their wording: “matched” samples between living and postmortem tissues. However, as they admitted in the discussion, living tissues were biased toward PD patients, and so on. When applicable, how about using “partially matched”?

Thank you for your comment. When applicable, we have modified the manuscript using the term ‘partially matched’ as well as ‘matched to the extent possible’.

2. I appreciated that the authors made it clear that the degree of editing level differences between living and postmortem tissues were not insignificant, especially in non-neuronal cell types. I suggest they emphasize this point while appreciating the utility of post-mortem tissues in future studies. For example, this point can be added to the last sentence of the introduction.

Thank you. We have clarified the last sentence to read as “Herein, we provide substantial evidence for significant differences in A-to-I editing profiles between postmortem and living human brain tissues, which are more evident in non-neuronal cell types (**Figure 1**).”

3. Regarding Figure 3B, although the volcano plot provides editing level differences with statistical significance, it will be helpful for readers to see the MA plot to see the relation between average editing levels and editing level differences.

Thank you for this suggestion. We now include a MA plot in **Supplemental Figure 7D**.

4. “A-to-G editing” has been used in the main text. It is advisable to use “A-to-I” for consistency.

We modified the manuscript accordingly and now use ‘A-to-I editing’ throughout the main text.

5. Many p-values were reported as values only: At least, the authors should mention what statistical tests were used and the number of samples used.

All statistical tests are described in each corresponding figure legend.

6. In the 6th paragraph of the discussion: “We also catalogued 2,362 context-dependent cis-edQTLs across 1,247 unique that differed between postmortem and living DLFPFC.” Does 1,247 indicate the number of loci?

Yes, 1,247 indicates loci and this is now clarified in the main text.